# Determinants of life satisfaction among women of reproductive age (15–49 years) in Bangladesh: A cross-sectional analysis

Nabil Ahmed Uthso, Noor Jahan Akter *

Institute of Statistical Research and Training, University of Dhaka, Dhaka, Bangladesh

☯ These authors contributed equally to this work.
* nakter1@isrt.ac.bd

**Data Availability Statement:** The data are available from the Multiple Indicator Cluster Survey (MICS) 2019 database which is available online. https://mics.unicef.org/surveys.

## Abstract

The life satisfaction of women is an essential component of their subjective well-being. It is an indicator of a woman's life quality based on personal perception. Considering the importance of women's subjective well-being, the United Nations (UN) has recognized this as one of its Sustainable Development Goals (SDGs). To the best of our knowledge, no study has been done yet for Bangladeshi women aged 15-49 years using nationally represented data. This study is the first study, to our knowledge, that will identify the determinants of life satisfaction and investigate the association between these determinants and life satisfaction among women of reproductive age (15-49 years) in Bangladesh. This study examined the overall life satisfaction and impact of some personal and sociodemographic characteristics on it among the women aged 15 to 49 years in Bangladesh using the Multiple Indicator Cluster Survey (MICS) 2019 data, a cross-sectional data. In our study, information regarding life satisfaction was available for a sample of 64,283 women after categorizing the variable overall life satisfaction and adjusting the missing values. The variable overall life satisfaction was categorized into three categories, namely low (0-3), moderate (4-6), and high (7-10), according to the Organisation for Economic Co-operation and Development (OECD) guidelines. Results showed that 14.67% of women are low satisfied, 50.65% are moderately satisfied, and 34.68% are highly satisfied with their lives. A bivariate analysis was applied in this study to show the significant association between the determinants and life satisfaction. The multinomial logistic regression analysis was performed to draw valid conclusions about the effects of the potential determinants on life satisfaction. The study revealed that satisfaction increases with age, education level, and wealth status. For the variable marital status, which has three categories: currently married, formerly married, and never married, current marriage was strongly associated with higher life satisfaction. In contrast, a former marriage and a never marriage were associated with lower life satisfaction. Migration status was also significantly associated with life satisfaction, where women who changed their location were more satisfied than those who never changed their current location. Bangladesh aims to achieve the Sustainable Development Goals (SDGs) by 2030. To achieve the Sustainable Development Goal (SDG) 3, which promotes well-being, and the goal 5, to empower all

**Funding:** The author(s) received no specific funding for this work.

**Competing interests:** The authors have declared that no competing interests exist.

women and girls, addressing the issues related to life satisfaction and applying appropriate intervention is a must.

## Introduction

Life satisfaction is an essential dimension of life. It is evaluated based on subjective assessment, which plays a crucial role in adolescent and adult women's lives [1]. Subjective perceptions of individuals of income, wealth, and health conditions significantly impact their lives, affecting their feeling of life satisfaction [2]. According to some researchers, life satisfaction depends on determinants such as socioeconomic status, health status, and social relationships that are non-material [3, 4]. In contrast, others postulate that life satisfaction is associated with factors like working status, actual income, and place of residence that are material [5, 6]. A society with less educated women has small incomes and reduced bargaining and decision-making power, and many women are not so satisfied with their lives. Moreover, the happiness and well-being of youth and adult people are strongly associated with life satisfaction [7, 8].

There are very few studies in developing countries regarding women's life satisfaction. Studies from developed countries found that income, health status, age, marital status, the highest level of education, and gender are significant factors associated with life satisfaction [5, 9–14]. A study showed that in eastern European countries, currently married, higher education level, increment in income, having no children, and rural residence were significantly associated with better life satisfaction and well-being [15].

Luhmann et al. [13] investigated that factors such as marriage and childbirth increase life satisfaction, whereas separation, migration, and losing a job decrease it. In developing countries, almost the same factors were significantly associated with life satisfaction. A study in Ghana showed a significant association between age, wealth index, gender, highest education level, marital status, type of residence, and marital status with life satisfaction [16]. Surprisingly, a study on Malawian people found no significant association between factors wealth index, education level with life satisfaction [17]. A study on Jamaican women found a significant association between factors such as level of education, wealth index, and marital status with life satisfaction but an insignificant association with place of residence [18].

Patton et al. [19] and Currie et al. [20] showed that subjective well-being is associated with health and behavior patterns. Physical health, mental health, and well-being trajectories are established in the youth and adolescence stages. Qasim et al. [21] found that women's well-being is significantly associated with early child development. In a previous study, Walker et al. [22] showed lower well-being; low satisfaction is significantly associated with low birth weight, stunting, decreasing children's survival, and many other complications. According to a study, the prevalence of underweight and stunting in Bangladesh is 36.2% and 41.1%, respectively (sample size = 7530), which is alarming [23].

Bangladesh, since its independence, has taken several steps to promote women's empowerment and increase the literacy rate, such as compulsory and free primary education programs, providing books free of cost, and stipends for female students. These steps are related to women's satisfaction and quality of life. However, in reality, many women are still unprivileged, do not get a quality education, and are deprived in many ways, which may reduce their satisfaction with their lives. Therefore, to attain the Sustainable Development Goals (SDGs) and to be a prosperous country, studying women's subjective well-being and life satisfaction is essential.

## Methodology

### Data source

In this study, we have used the Bangladesh Multiple Indicator Cluster Survey (MICS) 2019 data; that is, the data used in this research is secondary data. The Bangladesh MICS 2019 is a cross-sectional survey. With support from UNICEF, the Bangladesh Bureau of Statistics (BBS) conducted this observational study containing 140 indicators for women and children [24]. UNICEF developed this program in 1990 to support countries collecting data that includes several indicators of women and children. Each district's urban and rural areas were considered the primary sampling strata. First, a list of households was created within the chosen enumeration areas, and then 20 households were randomly drawn in each sample PSU using a systematic sampling method. Using the probability-proportion-to-size (PPS) method, these sample clusters were determined based on each cluster's total number of households. A total of 64,400 households in 3,220 sample clusters was chosen since the 2-stage stratified clustered sampling method was used. In this study, information regarding life satisfaction is available for a sample of 64,283 women after categorizing the variable overall life satisfaction and adjusting the missing values. The data is available on the MICS website free of cost (https://mics.unicef.org/surveys). Additionally, we have provided the data used in this study, the women's data, in the S1 File. The full data description and the detailed sampling procedure is available in the MICS 2019 report: *Progotir Pathey Bangladesh* [24].

### Ethical considerations

The BBS approved the survey protocol. In addition to verbal assent for each participant, adult consent was obtained for 15–17 years aged children before the children consent. The respondents provided information anonymously, knowing their participation's voluntary character, and acknowledged the confidentiality of the information. The respondents had the right to reject to answer any question. They also had the right to terminate the interview no matter when during the survey.

### Variables

**Outcome variable.** The dependent variable of this study is overall life satisfaction. The respondents were asked a question regarding their overall satisfaction with life. They were shown a picture of a ladder with steps with discrete values between 0 and 10. The respondents were asked to identify the ladder step that represents the satisfaction level with their lives. This variable has been categorized into three categories, namely, low satisfied (0–3), moderately satisfied (4–6), and highly satisfied (7–10), according to the Organisation for Economic Co-operation and Development (OECD) guidelines [2].

**Independent variables.** Based on previous literature, the independent variables were selected. The following variables are the independent variables of this study: area of residence (urban, rural), division (Dhaka, Barishal, Chattogram, Khulna, Mymensingh, Rajshahi, Rangpur, Sylhet), age (15–19, 20–24, 25–29, 30–34, 35–39, 40–44, 45–49 years), the highest education level (pre-primary or none, primary, secondary, higher secondary+), marital status (currently married, formerly married, never married), ethnicity (Bengali, others), functional disability (yes, no), wealth index (poorest, second quintile, middle, fourth quintile, richest), migration status (never moved, moved within the last 5 years, moved 5+ years ago).

## Statistical analysis

Univariate analyses were performed to show the distribution of women aged 15 to 49 years by their personal and sociodemographic characteristics. Bivariate analyses were used to observe the distribution of each determinant over the outcomes. The chi-square test assessed the significance of associations between the sociodemographic factors and life satisfaction. Since our response variable was a categorical variable with three categories, we performed a nominal logistic regression. The parallel regression or the proportionality of odds assumption was violated, making us unable to use proportional ordinal logistic regression. However, other ordinal regressions were not used due to a lack of understandable interpretation. The relative risk ratios (RRRs) were calculated, reported, and interpreted with a 95% confidence interval for each level of the determinants. The interpretation of RRR in the multinomial regression model has been described in the S3 File. The p-value threshold was chosen as 0.005, which Benjamin et al. [25] suggested. Hence, two-tailed statistical tests with p-value <0.005 were regarded statistically significant. All statistical analyses were performed, considering the complex survey design, as the dataset used in the study was derived from a multistage cluster survey. The statistical analyses were conducted with the help of the package "svy" in Stata (StataCorp version 14.0) software.

## Results

### Descriptive analysis

**Univariate analysis.** Table 1 shows that the overall percentage of low satisfied, moderately satisfied, and highly satisfied are 14.67%, 50.65%, and 34.68%, respectively. Most women are from rural areas between 15 and 19 years. The percentage of women completing higher secondary is 17.22%. The findings show that a large number of the women have met their secondary education, which is about 44%. Also, most of the women are currently married, which is about 80%. A very few women are formerly married, which is only 4.01% of the respondents. Moreover, most (98.83%) of the women are Bengali. The prevalence of functional disability is only 3.05%.

We see that the percentage of women increases with the wealth index. The percentage of poorest women is 17.50%, whereas 22.03% of women are from the wealthiest households. More than 31% of women never moved from their current location. In contrast, about 54% of respondents changed their location more than 5 years ago.

**Bivariate analysis.** Table 2 presents percentage distributions of life satisfaction levels for women. We observe that life satisfaction is low for about 15% of women, moderate for 50% of women, and 35% of women are highly satisfied with their lives. If we look area-wise, we see that percentage of moderately satisfied women is higher (51.79%) for rural women, while the percentage of moderately satisfied women is lower (46.94%) for urban women. The percentage of women with low life satisfaction is almost similar for urban and rural women, approximately 13% for urban, and 15% for rural. In Khulna, a small (9.14%) percentage of women have low life satisfaction, with the highest(44.36%) percentage of highly satisfied among all the divisions. The Mymenshing division has a large proportion of women who are moderately satisfied with their lives, about 62%.

The proportion of being moderately satisfied with life increases with age. For the age group 15–19 years, we see that the percentage of moderately satisfied women is about 46%, which is about 54% for 45–49 years. However, the percentage of highly satisfied women is high (41.10%) for the age group 15–19 years, and low (30.41%) for 40–44 years.

**Table 1. Distribution of reproductive (15–49 years) aged women by background characteristics.**

| Variables | Frequency | Percentage (%) |
|---|---|---|
| **Life satisfaction** | | |
| Low | 9,431 | 14.67 |
| Moderate | 32,559 | 50.65 |
| High | 22,293 | 34.68 |
| **Area** | | |
| Urban | 15,053 | 23.42 |
| Rural | 49,230 | 76.58 |
| **Division** | | |
| Barishal | 3,459 | 5.38 |
| Chattogram | 12,453 | 19.37 |
| Dhaka | 16,298 | 25.35 |
| Khulna | 7,577 | 11.79 |
| Mymenshing | 4,180 | 6.50 |
| Rajshahi | 8,517 | 13.25 |
| Rangpur | 7,081 | 11.02 |
| Sylhet | 4,717 | 7.34 |
| **Age (in years)** | | |
| 15–19 | 11,930 | 18.56 |
| 20–24 | 10,382 | 16.15 |
| 25–29 | 10,022 | 15.59 |
| 30–34 | 10,218 | 15.90 |
| 35–39 | 9,189 | 14.29 |
| 40–44 | 6,775 | 10.54 |
| 45–49 | 5,766 | 8.97 |
| **Highest education** | | |
| Pre-primary or none | 10,154 | 15.80 |
| Primary | 14,581 | 22.68 |
| Secondary | 28,477 | 44.30 |
| Higher secondary+ | 11,071 | 17.22 |
| **Marital status** | | |
| Currently married | 51,066 | 79.44 |
| Formerly married | 2,580 | 4.01 |
| Never married | 10,637 | 16.55 |
| **Functional disability** | | |
| Yes | 1,755 | 3.05 |
| No | 55,809 | 96.95 |
| **Ethnicity** | | |
| Bengali | 63,534 | 98.83 |
| Others | 749 | 1.17 |
| **Wealth status** | | |
| Poorest | 11,248 | 17.50 |
| Second quintile | 12,312 | 19.15 |
| Middle | 12,965 | 20.17 |
| Fourth quintile | 13,598 | 21.15 |
| Richest | 14,160 | 22.03 |
| **Migration** | | |
| Never moved | 20,226 | 31.47 |

(*Continued*)

**Table 1.** (Continued)

| Variables | Frequency | Percentage (%) |
|-----------|-----------|----------------|
| Within the last 5 years | 9,588 | 14.92 |
| 5+ years ago | 34,458 | 53.61 |

The frequency table shows that the proportion of highly satisfied women constantly increases with education. About 52% of women are highly satisfied with their lives who have completed their higher secondary education. The women with education level pre-primary or none have the lowest (.19) probability of being highly satisfied and highest (.25) probability of having low satisfaction. The higher secondary+ category has the most minuscule (6.97%) percentage of women with low life satisfaction.

About 42% of women are highly satisfied with their lives who are never married, whereas only 12.40% of formerly married women have high life satisfaction. Approximately 52% of the currently married women are moderately satisfied with their lives. The percentage of moderately satisfied women decreases from currently married to never married. Economic status is also associated with the level of life satisfaction. The percentage of highly satisfied women increases from 18.35% to 52.22% with economic status. The poorest group has the highest (26.01%) percentage of low satisfied women, while this percentage is lowest (7.07%) for the wealthiest group. Moreover, about 53% of women who migrated more than 5 years ago are moderately satisfied. The percentage of moderately satisfied women is 47.79%, who never moved from their present location.

## Multinomial logistic regression

**Moderate satisfaction relative to low satisfaction.** From Table 3, we see that for rural relative to urban, the relative risk of having moderate life satisfaction relative to low life satisfaction is expected to increase by 26% (p-value <0.001), holding other variables constant. In other words, rural women are more likely than urban women to have moderate life satisfaction than low life satisfaction. For Chattogram division relative to Barishal, the relative risk of having moderate life satisfaction relative to low life satisfaction is expected to decrease by 38% (p-value <0.001) holding other variables constant. Women of the Chattogram division are less likely than women of the Barishal division to have moderate life satisfaction than low life satisfaction. Also, the women of the Mymenshing division have 66% (p-value <0.001) more relative risk than women of the Barishal division to have moderate life satisfaction relative to low life satisfaction. Moreover, women of the Dhaka division relative to Barishal have 24% (p-value <0.001) less relative risk to have moderate life satisfaction than low life satisfaction.

For the age-predicted life satisfaction, women aged 45–49 years have 34% (p-value <0.001) more relative risk to be moderately satisfied than low relative to women aged 15–19 years. For higher secondary passed women than uneducated women, the relative risk of having moderate life satisfaction relative to low life satisfaction increases by a factor of 2.39, holding other variables constant (p-value <0.001). In other words, highly educated women are more likely than uneducated women to have moderate life satisfaction than low life satisfaction. For marital status, formerly married women have 63% (p-value: 0.003) less relative risk to be highly satisfied than low relative to women who are currently married. However, women who never married are 21% (p-value <0.001) less likely than currently married women to have moderate life satisfaction relative to low satisfaction.

Women with no functional disability are 41% (p-value <0.001) more likely than women with disability to have moderate life satisfaction than low life satisfaction. Non-Bengali women

**Table 2. Results of bivariate analysis representing association between determinants with life satisfaction with p-value.**

| Variables | % Low satisfied | % Moderately satisfied | % Highly satisfied | N | p-value |
|---|---|---|---|---|---|
| Overall | 14.67 | 50.65 | 34.68 | 64,283 | |
| **Area** | | | | | <0.001 |
| Urban | 12.71 | 46.94 | 40.35 | 15,053 | |
| Rural | 15.27 | 51.79 | 32.94 | 49,230 | |
| **Division** | | | | | < 0.001 |
| Barishal | 15.57 | 53.35 | 31.08 | 3,459 | |
| Chattogram | 16.44 | 42.17 | 41.38 | 12,453 | |
| Dhaka | 14.14 | 48.58 | 37.28 | 16,298 | |
| Khulna | 9.14 | 46.49 | 44.36 | 7,577 | |
| Mymenshing | 11.28 | 61.78 | 26.94 | 4,180 | |
| Rajshahi | 17.07 | 55.03 | 27.90 | 8,517 | |
| Rangpur | 16.01 | 56.56 | 27.43 | 7,081 | |
| Sylhet | 16.77 | 58.21 | 25.03 | 4,717 | |
| **Age (in years)** | | | | | <0.001 |
| 15–19 | 13.21 | 45.69 | 41.10 | 11,930 | |
| 20–24 | 12.87 | 49.09 | 38.04 | 10,382 | |
| 25–29 | 14.42 | 50.84 | 34.74 | 10,022 | |
| 30–34 | 15.45 | 52.68 | 31.87 | 10,218 | |
| 35–39 | 16.67 | 52.46 | 30.87 | 9,189 | |
| 40–44 | 16.18 | 53.40 | 30.41 | 6,775 | |
| 45–49 | 15.06 | 53.67 | 31.27 | 5,766 | |
| **Highest education** | | | | | <0.001 |
| Pre-primary or none | 24.97 | 56.44 | 18.59 | 10,154 | |
| Primary | 19.03 | 56.08 | 24.89 | 14,581 | |
| Secondary | 11.77 | 49.38 | 38.86 | 28,477 | |
| Higher secondary+ | 6.97 | 41.47 | 51.56 | 11,071 | |
| **Marital status** | | | | | <0.001 |
| Currently married | 13.79 | 52.01 | 34.20 | 51,066 | |
| Formerly married | 38.32 | 49.29 | 12.40 | 2,580 | |
| Never married | 13.17 | 44.45 | 42.38 | 10,637 | |
| **Functional disability** | | | | | <0.001 |
| Yes | 22.31 | 50.64 | 27.05 | 1,755 | |
| No | 14.61 | 51.30 | 34.09 | 55,809 | |
| **Ethnicity** | | | | | 0.011 |
| Bengali | 14.66 | 50.58 | 34.76 | 63,534 | |
| Others | 15.67 | 56.53 | 27.81 | 749 | |
| **Wealth index** | | | | | <0.001 |
| Poorest | 26.01 | 55.64 | 18.35 | 11,248 | |
| Second quintile | 18.83 | 57.60 | 23.57 | 12,312 | |
| Middle | 13.27 | 53.52 | 33.22 | 12,965 | |
| Fourth quintile | 10.78 | 47.86 | 41.36 | 13,598 | |
| Richest | 7.07 | 40.07 | 52.22 | 14,160 | |
| **Migration** | | | | | <0.001 |
| Never moved | 17.47 | 47.79 | 34.74 | 20,226 | |
| Within the last 5 years | 12.23 | 47.82 | 39.95 | 9,588 | |
| 5+ years ago | 13.70 | 53.12 | 33.17 | 34,458 | |

**Table 3. Results of multinomial logistic regression model regressing on life satisfaction showing relative risk ratios (RRRs), 95% confidence interval, and p-value.**

| Variables | Moderate satisfaction relative to low | | High satisfaction relative to low | |
|---|---|---|---|---|
| | RRR (95% CI) | p-value | RRR (95% CI) | p-value |
| **Area** | | | | |
| Urban | (ref.) | | | |
| Rural | 1.26 (1.18, 1.35) | <0.001 | 1.56 (1.44, 1.68) | <0.001 |
| **Division** | | | | |
| Barishal | (ref.) | | | |
| Chattogram | 0.62 (0.55, 0.69) | <0.001 | 0.78 (0.68, 0.89) | <0.001 |
| Dhaka | 0.76 (0.67, 0.85) | <0.001 | 0.69 (0.61, 0.79) | <0.001 |
| Khulna | 1.21 (1.05, 1.38) | 0.006 | 1.61 (1.39, 1.86) | <0.001 |
| Mymenshing | 1.66 (1.43, 1.93) | <0.001 | 1.15 (0.97, 1.36) | 0.106 |
| Rajshahi | 0.83 (0.73, 0.94) | 0.002 | 0.64 (0.56, 0.74) | <0.001 |
| Rangpur | 0.97 (0.86, 1.10) | 0.650 | 0.77 (0.67, 0.89) | <0.001 |
| Sylhet | 0.91 (0.79, 1.05) | 0.191 | 0.49 (0.42, 0.57) | <0.001 |
| **Age (in years)** | | | | |
| 15–19 | (ref.) | | | |
| 20–24 | 1.00 (0.89, 1.12) | 0.989 | 0.88 (0.78, 0.99) | 0.030 |
| 25–29 | 0.94 (0.84, 1.06) | 0.322 | 0.79 (0.70, 0.91) | 0.001 |
| 30–34 | 0.96 (0.85, 1.08) | 0.457 | 0.78 (0.68, 0.88) | <0.001 |
| 35–39 | 0.98 (0.86, 1.11) | 0.707 | 0.83 (0.73, 0.95) | 0.008 |
| 40–44 | 1.12 (0.98, 1.28) | 0.093 | 1.01 (0.87, 1.17) | 0.898 |
| 45–49 | 1.35 (1.18, 1.55) | <0.001 | 1.39 (1.19, 1.62) | <0.001 |
| **Highest education** | | | | |
| Pre-primary or none | (ref.) | | | |
| Primary | 1.29 (1.21, 1.39) | <0.001 | 1.59 (1.47, 1.74) | <0.001 |
| Secondary | 1.69 (1.57, 1.83) | <0.001 | 2.94 (2.69, 3.21) | <0.001 |
| Higher secondary+ | 2.39 (2.13, 2.67) | <0.001 | 5.79 (5.12, 6.54) | <0.001 |
| **Marital status** | | | | |
| Currently married | (ref.) | | | |
| Formerly married | 0.37 (0.34, 0.41) | 0.003 | 0.15 (0.13, 0.17) | <0.001 |
| Never married | 0.79 (0.71, 0.89) | <0.001 | 0.83(0.73, 0.94) | 0.003 |
| **Functional disability** | | | | |
| Yes | (ref.) | | | |
| No | 1.41 (1.24, 1.61) | <0.001 | 1.59 (1.37, 1.86) | <0.001 |
| **Ethnicity** | | | | |
| Bengali | (ref.) | | | |
| Others | 2.30 (1.83, 2.89) | <0.001 | 2.12 (1.63, 2.75) | <0.001 |
| **Wealth index** | | | | |
| Poorest | (ref.) | | | |
| Second quintile | 1.37 (1.28, 1.47) | <0.001 | 1.59 (1.46, 1.74) | <0.001 |
| Middle | 1.87 (1.73, 2.02) | <0.001 | 3.05 (2.78, 3.33) | <0.001 |
| Fourth quintile | 2.11 (1.94, 2.29) | <0.001 | 4.69 (4.26, 5.16) | <0.001 |
| Richest | 2.86 (2.57, 3.18) | <0.001 | 9.29 (8.28, 10.4) | <0.001 |
| **Migration** | | | | |
| Never moved | (ref.) | | | |
| Within the last 5 years | 1.15 (1.05, 1.26) | <0.001 | 1.32 (1.19, 1.46) | <0.001 |
| 5+ years ago | 1.39 (1.31, 1.48) | <0.001 | 1.56 (1.46, 1.68) | <0.001 |

are 2.30 times more likely than Bengali women to have moderate life satisfaction than low life satisfaction (p-value <0.001). Furthermore, as the wealth index increases, the relative risk ratio increases. Richest women are 2.86 times more likely than the poorest women to have moderate life satisfaction than low (p-value <0.001). Moreover, Women who moved within the last 5 years are 15% (p-value <0.001) more likely than women who never moved from their present location to be moderately satisfied with their life than low satisfaction. Also, women who moved more than 5 years ago are 39% (p-value <0.001) more likely than women who never moved to have moderate life satisfaction than low.

**High satisfaction relative to low satisfaction.**   For rural relative to urban, the relative risk of having high life satisfaction relative to low life satisfaction is expected to increase by 56% (p-value <0.001), holding other variables constant. Rural women are more likely than urban women to have high life satisfaction than low life satisfaction. For Chattogram division relative to Barishal, the relative risk of having high life satisfaction relative to low life satisfaction is expected to decrease by 22% (p-value <0.001), holding other variables constant. Women of the Chattogram division are less likely than those of the Barishal division to have high life satisfaction than low life satisfaction. Also, the women of the Khulna division are 61% (p-value <0.001) more likely than women of the Barishal division to have high life satisfaction than low life satisfaction. Moreover, women of the Dhaka division relative to Barishal are 31% (p-value <0.001) less likely to have high life satisfaction than low life satisfaction.

For the age-predicted life satisfaction, women aged 45–49 years are 39% (p-value <0.001) more likely than women of 15–19 years to have high life satisfaction than low life satisfaction. For higher secondary-passed women relative to uneducated women, the relative risk of having high life satisfaction relative to low life satisfaction increases by 5.79 times, holding other variables constant (p-value <0.001). In other words, highly educated women are more likely to have high life satisfaction than low life satisfaction. For marital status, formerly married women are 85% (p-value <0.001) less likely than women who are currently married to be highly satisfied than low. However, women who never married are 17% (p-value: 0.003) less likely than currently married women to have high life satisfaction than low.

Women with no functional disability are 59% (p-value <0.001) more likely than those with a disability to have high life satisfaction than low life satisfaction. Non-Bengali women are 2.12 times more likely than Bengali women to have high life satisfaction than low life satisfaction (p-value <0.001). Furthermore, as the wealth index increases, the relative risk ratio increases. Richest women are 9.29 times more likely than the poorest women to have high life satisfaction than low (p-value <0.001). Moreover, Women who moved within the last 5 years are 32% (p-value <0.001) more likely than women who never moved from their present location to be highly satisfied with their life than low satisfaction. Also, women who moved more than 5 years ago are 56% (p-value <0.001) more likely than women who never moved to have high life satisfaction than low.

## Discussion

This study examines overall life satisfaction and its potential determinants for Bangladeshi reproductive-aged women. Our findings show that living in rural areas is positively associated with women's life satisfaction, which contradicts the study of Wang et al. [26], but is consistent with the study of Cappa et al. [15]. The reason may be because urban women have to deal with traffic jams, air pollution, sound pollution, and many other troublesome things in their daily life, which may reduce their satisfaction with life. Upadyaya et al. [27] found that people who live in areas with huge populations have reduced life satisfaction.

Within the division groups, those who live in Chattogram, Dhaka, and Rajshahi have a reduced relative risk of being moderately and highly satisfied than those who live in Barishal. According to the SVRS 2020 [28], the literacy rate of women (15+ years) in Barishal is 82.5%, the highest in the country. Since the literacy rate is closely related to awareness about rights and education, it may be related to higher life satisfaction for women in the Barishal division. We find that the elderly adults are more satisfied as the transformation from youth to adult is a crucial period associated with various responsibilities, including achieving life goals like getting a higher education, getting a job, and getting married [29]. Fulfilling these goals puts pressure and stress on young women, which reduces their well-being and satisfaction with their lives [30, 31].

We find that being highly educated is related to higher life satisfaction for women, which agrees with McIntyre et al. [18] and Powdthavee et al. [32], who also found that being highly educated is significantly associated with higher satisfaction levels. Riddell et al. [33] stated that education creates better opportunities and reduces unemployment in the society. Therefore, job opportunities indirectly serve an essential role in subjective life satisfaction. Moreover, women who are more educated have higher incomes than women who are less educated or uneducated. As a result, women of higher income are more likely to lead an enjoyable and luxurious life [18]. On the other hand, less educated women struggle to get a job and usually have little earnings to maintain their lives, decreasing their satisfaction.

We also find that currently married women are more satisfied with their lives than those formerly married and never married. This finding contradicts the results of McIntyre et al. [18] and Addai et al. [34], who found that females who were never married were more satisfied than currently and formerly married women. However, our finding is consistent with the results of Owusu et al. [16], Yaya et al. [17] and Botha et al. [35], who also found that married women have higher satisfaction than formerly married women. There may be a reason women previously married and never married undergo constant criticism and bullying from society, hence reduced satisfaction with life. According to the study of Næss et al. [36], marriage is all about affection, supporting each other, safety, financial security, and many other benefits that increase one's life satisfaction.

Our results show that women who have any functional disability are less satisfied with their lives than those who have no disability, which is consistent with findings of Addabbo et al. [37]. In terms of ethnicity, Bengali women are less likely to be happy with their lives than non-Bengali women. Further studies are needed to assess whether reporting low life satisfaction of Bengali women is due to their societal culture, conservative family mindset, or ignorance about their rights. Many other studies have found a significant association between ethnicity and life satisfaction [38, 39].

Finally, this research reveals that life satisfaction increases with a woman's wealth or income. Women from wealthy households can meet their needs and are not so much suffered from financial burdens. According to the research of Kreager et al. [40], money gives women more freedom and independence to make family decisions and increases bargaining power. Jan and Masood [41] stated that the socio-economic status of women has a profound impact on their life satisfaction. In our study, we also include an independent variable: migration. Results show that women who never changed their present location are less satisfied than those who changed their location.

## Strengths and limitations

This study's strength is its ability to generate valuable information regarding the potential factors of life satisfaction among reproductive-aged women in Bangladesh. Also, this study used a

nationally represented dataset that increases reliability by reducing the effect of possible errors. Despite being a nationally representative dataset, the study's cross-sectional nature does not allow us to establish causation between the determinants and life satisfaction. Other factors previously associated with women's life satisfaction that we can not measure due to the unavailability of information, such as smoking status and alcohol consumption, are also limitations of this study. Moreover, the data suffer from a small number of missing values.

## Conclusion

This study's findings suggest that place of residence, the divisional region, respondent's age, education level, marital status, functional disability, ethnicity, wealth status, and migration status are the potential determinants of life satisfaction among Bangladeshi women of reproductive age. We recommend that researchers conduct more studies regarding life satisfaction for Bangladeshi people to understand their challenges so that we can have a better society. Since the association between sociodemographic characteristics and satisfaction with life is crucial, improving the socioeconomic status and addressing the urban-rural gap in well-being is crucial to promoting the advanced quality of life among Bangladeshi women. Further in-depth surveys and studies are required to understand better the association between the divisional region and satisfaction with life, life satisfaction and literacy rate, rural residence and increased life satisfaction, and why life satisfaction increases with migration. Continuing to collect these from Bangladeshi women, we highly recommend collecting information on subjective well-being from Bangladeshi men for better comparison.

## Supporting information

**S1 File. Stata dataset of the study.** Stata dataset containing information of reproductive (15–49 years) aged Bangladeshi women from MICS 2019.
(DTA)

**S2 File. Stata do-file of the study.** Stata do-file that contains the stata codes for the statistical analysis.
(DO)

**S3 File. Multinomial logistic regression and RRR.** Definition of multinomial logistic regression and interpretation of relative risk ratio (RRR).
(ZIP)

## Acknowledgments

We want to thank the Multiple Indicator Cluster Survey (MICS) authority for making the data available free of cost. We want to also thank the Bangladesh Bureau of Statistics (BBS) and UNICEF Bangladesh for carrying out the survey.

## Author Contributions

**Conceptualization:** Noor Jahan Akter.

**Data curation:** Nabil Ahmed Uthso.

**Formal analysis:** Nabil Ahmed Uthso.

**Methodology:** Nabil Ahmed Uthso, Noor Jahan Akter.

**Supervision:** Noor Jahan Akter.

**Writing – original draft:** Nabil Ahmed Uthso.

**Writing – review & editing:** Nabil Ahmed Uthso, Noor Jahan Akter.

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
