## [Decision Letter · Decision Letter 0]

4 Aug 2022

PONE-D-22-19296Determinants of Life Satisfaction among Women of Reproductive Age (15-49 years) in Bangladesh: How much

Bangladeshi Women are Satisfied with their Lives?PLOS ONE

Dear Dr. Akter,

Thank you for submitting your manuscript to PLOS ONE. After careful consideration, we feel that it has merit but does not fully meet PLOS ONE’s publication criteria as it currently stands. Therefore, we invite you to submit a revised version of the manuscript that addresses the points raised during the review process.

We look forward to receiving your revised manuscript.

Kind regards,

Tariq Jamal Siddiqi

Academic Editor

PLOS ONE

Journal Requirements:

" ext-link-type="uri" xlink:type="simple">https://journals.plos.org/plosone/s/file?id=ba62/PLOSOne_formatting_sample_title_authors_affiliations.pdf"

2. In the ethics statement in the Methods and online submission information, please ensure that you have specified what type you obtained (for instance, written or verbal, and if verbal, how it was documented and witnessed). If your study included minors, state whether you obtained consent from parents or guardians. If the need for consent was waived by the ethics committee, please include this information.

3. We noted in your submission details that a portion of your manuscript may have been presented or published elsewhere. 

"It is a secondary data of Multiple Indicator Cluster Survey (MICS) 2019"

4. Please amend your authorship list in your manuscript file to include authors Noor Jahan Akter and Nabil Ahmed Uthso.

5. Please upload a copy of Supporting Information File S1 and S2 which you refer to in your text.

Reviewers' comments:

Reviewer's Responses to Questions

**Comments to the Author**

1. Is the manuscript technically sound, and do the data support the conclusions?

Reviewer #1: Yes

2. Has the statistical analysis been performed appropriately and rigorously? 

Reviewer #1: Yes

3. Have the authors made all data underlying the findings in their manuscript fully available?

Reviewer #1: Yes

4. Is the manuscript presented in an intelligible fashion and written in standard English?

Reviewer #1: Yes

5. Review Comments to the Author

Reviewer #1: The authors conducted a study using cross-sectional data from the Multiple Indicator Cluster Survey 2019 to identify and analyze the determinants of life satisfaction among women of reproductive age (15 - 49 years) in Bangladesh. Although the manuscript is well-drafted and data is presented in an intelligible manner, there are concerns that need to be addressed in order to make the article worthy of consideration.

1. The title of the article has repetition. The latter question 'How much Bangladeshi Women are Satisfied with their Lives?' is reiterating the same idea mentioned in the former statement. The authors should consider removing this part and replacing it with the key methodology of the paper, for instance 'Determinants of Life Satisfaction among Women of Reproductive Age (15-49 years) in Bangladesh: A Cross-Sectional Analysis' or 'Determinants of Life Satisfaction among Women of Reproductive Age (15-49 years) in Bangladesh: A Retrospective Cross-Sectional Study' or something of the sort.

2. In the abstract, the authors say 'The migration status was found to be positively associated with higher life satisfaction '. This sentence is vague and does not hold much meaning. Kindly elucidate upon the migration status - which migration status led to a more satisfied life and which did not?

3. The introduction is pretty long and does not highlight the rationale of this article clearly. Kindly shorten it by removing less important details and ensure that the last paragraph of introduction specifically highlights the need for this study to be conducted, in other words, builds up strongly on the rationale.

4. In the methods, the sections entitled 'Nominal logistic regression' and 'Relative risk ratio (RRR) in multinomial logistic regression' do not need to be mentioned in the main text of the article. Kindly remove them and place them as supplementary material.

5. Can the authors please state the company details of Stata software used within brackets.

6. Statistical significance with reported p-values is an integral part of scientific research. The authors need to acknowledge what value of p was considered significant in their study in the statistical analysis section within the methods. Also kindly state the p-values in brackets within the text while stating the results as well.

7. The authors need to mention the design of their study clearly within the methods - retrospective or observational etc.

8. The authors state 'Conclusion: there may be a reason that women who are formerly married and never married undergo constant criticism and bullying from society'. Kindly elaborate upon this statement. Also please remove the word 'conclusion' from here since it should not be mentioned this way among the main text of discussion.

9. Throughout the discussion, the authors need to ensure that they are not constantly regurgitating the results or paraphrasing their findings. It is of utmost importance to 'discuss' the results, contrast and compare them. Make sure to provide valuable future implications for your article.

10. Kindly move the strengths and limitations section before the conclusion.

11. The conclusion is extremely long. Kindly cut it short to no more than 4 to 5 sentences.

12. Names of authors and details in the author contribution sections are missing.

6. PLOS authors have the option to publish the peer review history of their article (what does this mean?). If published, this will include your full peer review and any attached files.

Reviewer #1: No

---

## [Author Response · Author response to Decision Letter 0]

15 Sep 2022

Rebuttal letter

We thank the editor and the reviewer for their comments on our manuscript. Below is our response to each point raised by the academic editor and reviewer. We hope that we satisfyingly addressed them and that the manuscript will be now suited for publication.

Sincerely,

On behalf of all authors,

Noor Jahan Akter

Academic editor:

Response: Included.

Response: Included.

Response: Included.

and 

https://journals.plos.org/plosone/s/file?id=ba62/PLOSOne_formatting_sample_title_authors_affiliations.pdf"

Response: Thank you. We have tried to make our revised manuscript meet the PLOS ONE’s style requirements.

2. In the ethics statement in the Methods and online submission information, please ensure that you have specified what type you obtained (for instance, written or verbal, and if verbal, how it was documented and witnessed). If your study included minors, state whether you obtained consent from parents or guardians. If the need for consent was waived by the ethics committee, please include this information.

Response: Since we have used secondary data, we did not conduct the survey. The Bangladesh Bureau of Statistics (BBS) and UNICEF conducted the Multiple Indicator Cluster Survey (MICS) 2019. Therefore, the BBS and UNICEF obtained ethical clearance. However, the issues you have raised have been addressed in the ethical considerations section of the revised manuscript and the online submission system.

3. We noted in your submission details that a portion of your manuscript may have been presented or published elsewhere. 

"It is a secondary data of Multiple Indicator Cluster Survey (MICS) 2019"

Response: No part of this manuscript was peer-reviewed or published in any journal.

4. Please amend your authorship list in your manuscript file to include authors Noor Jahan Akter and Nabil Ahmed Uthso.

Response: Thank you for pointing this out. Authors’ names have been included in the revised manuscript.

5. Please upload a copy of Supporting Information File S1 and S2 which you refer to in your text.

Response: Supporting information files have been uploaded.

Reviewers' comments:

Reviewer's Responses to Questions

Comments to the Author

1. Is the manuscript technically sound, and do the data support the conclusions?

Reviewer #1: Yes

2. Has the statistical analysis been performed appropriately and rigorously?

Reviewer #1: Yes

3. Have the authors made all data underlying the findings in their manuscript fully available?

Reviewer #1: Yes

4. Is the manuscript presented in an intelligible fashion and written in standard English?

Reviewer #1: Yes

5. Review Comments to the Author

Reviewer #1: The authors conducted a study using cross-sectional data from the Multiple Indicator Cluster Survey 2019 to identify and analyze the determinants of life satisfaction among women of reproductive age (15 - 49 years) in Bangladesh. Although the manuscript is well-drafted and data is presented in an intelligible manner, there are concerns that need to be addressed in order to make the article worthy of consideration.

1. The title of the article has repetition. The latter question 'How much Bangladeshi Women are Satisfied with their Lives?' is reiterating the same idea mentioned in the former statement. The authors should consider removing this part and replacing it with the key methodology of the paper, for instance 'Determinants of Life Satisfaction among Women of Reproductive Age (15-49 years) in Bangladesh: A Cross-Sectional Analysis' or 'Determinants of Life Satisfaction among Women of Reproductive Age (15-49 years) in Bangladesh: A Retrospective Cross-Sectional Study' or something of the sort.

Response: Thank you for this suggestion. The title has been changed to ‘Determinants of Life Satisfaction among Women of Reproductive Age (15-49 years) in Bangladesh: A Cross-Sectional Analysis’ according to your suggestion.

2. In the abstract, the authors say 'The migration status was found to be positively associated with higher life satisfaction '. This sentence is vague and does not hold much meaning. Kindly elucidate upon the migration status - which migration status led to a more satisfied life and which did not?

Response: Thank you for pointing that out. The sentence has been changed to ‘Migration status was also significantly associate with life satisfaction, where women who changed their location were more satisfied than those who never changed their current location’ in the abstract.

3. The introduction is pretty long and does not highlight the rationale of this article clearly. Kindly shorten it by removing less important details and ensure that the last paragraph of introduction specifically highlights the need for this study to be conducted, in other words, builds up strongly on the rationale.

Response: Thank you for bringing this issue to us. The introduction has been revised.

4. In the methods, the sections entitled 'Nominal logistic regression' and 'Relative risk ratio (RRR) in multinomial logistic regression' do not need to be mentioned in the main text of the article. Kindly remove them and place them as supplementary material.

Response: The sections ‘Nominal logistic regression’ and ‘Relative risk ratio (RRR) in multinomial logistic regression’ have been placed as supplementary material (S3 File) according to your suggestion. 

5. Can the authors please state the company details of Stata software used within brackets.

Response: Thank you for pointing that out. Company details of Stata software have been stated within brackets as ‘Stata (StataCorp version 14.0)’ in the statistical analysis section (line 108).

6. Statistical significance with reported p-values is an integral part of scientific research. The authors need to acknowledge what value of p was considered significant in their study in the statistical analysis section within the methods. Also kindly state the p-values in brackets within the text while stating the results as well.

Response: Thank you for raising this issue. The p-value threshold has been added in the statistical analysis section (line 104), and p-values have been stated in brackets within the text. 

7. The authors need to mention the design of their study clearly within the methods - retrospective or observational etc.

Response: The study’s design has been mentioned in the data source subsection of the methodology section (line 50-51).

8. The authors state 'Conclusion: there may be a reason that women who are formerly married and never married undergo constant criticism and bullying from society'. Kindly elaborate upon this statement. Also please remove the word 'conclusion' from here since it should not be mentioned this way among the main text of discussion.

Response: The word ‘conclusion’ has been removed in the revised manuscript (line 257-258).

9. Throughout the discussion, the authors need to ensure that they are not constantly regurgitating the results or paraphrasing their findings. It is of utmost importance to 'discuss' the results, contrast and compare them. Make sure to provide valuable future implications for your article.

Response: Thank you for pointing that out. The discussion section has been revised according to your suggestion.

10. Kindly move the strengths and limitations section before the conclusion.

Response: The section ‘Strengths and limitations’ has been moved before the conclusion section as per your advice.

11. The conclusion is extremely long. Kindly cut it short to no more than 4 to 5 sentences. 

Response: Thank you for this suggestion. The conclusion has been shortened into 5 sentences according to your advice.

12. Names of authors and details in the author contribution sections are missing.

Response: The authors’ name and author contributions section has been added to the revised manuscript.

We humbly thank the editor and the reviewer for their wise suggestions and advice to improve the manuscript.

---

## [Decision Letter · Decision Letter 1]

10 Oct 2022

Determinants of Life Satisfaction among Women of Reproductive Age (15-49 years) in Bangladesh: A Cross-Sectional Analysis

PONE-D-22-19296R1

Dear Dr. Akter,

We’re pleased to inform you that your manuscript has been judged scientifically suitable for publication and will be formally accepted for publication once it meets all outstanding technical requirements.

Kind regards,

Tariq Jamal Siddiqi

Academic Editor

PLOS ONE

Additional Editor Comments (optional):

Reviewers' comments:

Reviewer's Responses to Questions

**Comments to the Author**

1. If the authors have adequately addressed your comments raised in a previous round of review and you feel that this manuscript is now acceptable for publication, you may indicate that here to bypass the “Comments to the Author” section, enter your conflict of interest statement in the “Confidential to Editor” section, and submit your "Accept" recommendation.

Reviewer #1: All comments have been addressed

2. Is the manuscript technically sound, and do the data support the conclusions?

Reviewer #1: Yes

3. Has the statistical analysis been performed appropriately and rigorously? 

Reviewer #1: Yes

4. Have the authors made all data underlying the findings in their manuscript fully available?

Reviewer #1: Yes

5. Is the manuscript presented in an intelligible fashion and written in standard English?

Reviewer #1: Yes

6. Review Comments to the Author

Reviewer #1: (No Response)

7. PLOS authors have the option to publish the peer review history of their article (what does this mean?). If published, this will include your full peer review and any attached files.

Reviewer #1: No

---

## [Editor Report · Acceptance letter]

21 Oct 2022

PONE-D-22-19296R1 

Determinants of Life Satisfaction among Women of Reproductive Age (15-49 years) in Bangladesh: A Cross-Sectional Analysis 

Dear Dr. Akter:

I'm pleased to inform you that your manuscript has been deemed suitable for publication in PLOS ONE. Congratulations! Your manuscript is now with our production department. 

Kind regards, 

on behalf of

Dr. Tariq Jamal Siddiqi 

Academic Editor

PLOS ONE